**Data Availability Statement:** All data are uploaded to the EPFL-LNCO database and accessible via the following URL: https://gitlab.epfl.ch/lnco-public/progin2020.

# Somatosensory-visual effects in visual biological motion perception

Pierre Progin[1,2,3]*, Nathan Faivre[1,3,4], Anna Brooks[5,6], Wenwen Chang[7], Manuel Mercier[8], Lars Schwabe[9], Kim Q. Do[10,11], Olaf Blanke[2,3,11,12]*

1 Department of Psychiatry, Service of General Psychiatry, Lausanne University Hospital (CHUV), Lausanne, Switzerland, 2 Laboratory of Cognitive Neuroscience, Brain Mind Institute, Faculty of Life Sciences, Swiss Federal Institute of Technology, Geneva, Switzerland, 3 Center for Neuroprosthetics, Faculty of Life Sciences, Swiss Federal Institute of Technology, Geneva, Switzerland, 4 CNRS, LPNC UMR 5105, Université Grenoble Alpes, Grenoble, France, 5 Lifeline Research Foundation, Lifeline Australia, Deakin ACT, Australia, 6 School of Health and Human Sciences, Southern Cross University, Lismore NSW, Australia, 7 Department of Mechanical Engineering and Automation, Northeastern University, Shenyang, China, 8 Institut de Neurosciences des Systèmes (INS), Inserm (U1106), Aix Marseille University, Marseille, France, 9 Data Analytics, Artificial Intelligence and Blockchain, Lufthansa Industry Solutions AS, Norderstedt, Germany, 10 Department of Psychiatry, Center for Psychiatric Neuroscience, Lausanne University Hospital (CHUV), Lausanne, Switzerland, 11 National Center of Competence in Research (NCCR) "SYNAPSY—The Synaptic Bases of Mental Diseases", Lausanne, Switzerland, 12 Department of Neurology, University Hospital Geneva, Geneva, Switzerland

* pierre.progin@chuv.ch (PP); olaf.blanke@epfl.ch (OB)

## Abstract

Social cognition is dependent on the ability to extract information from human stimuli. Of those, patterns of biological motion (BM) and in particular walking patterns of other humans, are prime examples. Although most often tested in isolation, BM outside the laboratory is often associated with multisensory cues (i.e. we often hear and see someone walking) and there is evidence that vision-based judgments of BM stimuli are systematically influenced by motor signals. Furthermore, cross-modal visuo-tactile mechanisms have been shown to influence perception of bodily stimuli. Based on these observations, we here investigated if somatosensory inputs would affect visual BM perception. In two experiments, we asked healthy participants to perform a speed discrimination task on two point light walkers (PLW) presented one after the other. In the first experiment, we quantified somatosensory-visual interactions by presenting PLW together with tactile stimuli either on the participants' fore-arms or feet soles. In the second experiment, we assessed the specificity of these interactions by presenting tactile stimuli either synchronously or asynchronously with upright or inverted PLW. Our results confirm that somatosensory input in the form of tactile foot stimulation influences visual BM perception. When presented with a seen walker's footsteps, additional tactile cues enhanced sensitivity on a speed discrimination task, but only if the tactile stimuli were presented on the relevant body-part (under the feet) and when the tactile stimuli were presented synchronously with the seen footsteps of the PLW, whether upright or inverted. Based on these findings we discuss potential mechanisms of somatosensory-visual interactions in BM perception.

**Funding:** Authors were supported by funds from the Swiss Federal Institute of Technology (EPFL). The funder had no role in study design, data collection and analysis, decision to publish, or preparation of the manuscript.

**Competing interests:** The authors have declared that no competing interests exist.

## Introduction

Actions of other individuals are a fundamental source of information for social agents and as such, the ability to exploit the sensory information they generate is important for social cognition. Research has focused on the ability of human observers to interpret biological motion (hereafter BM) of their conspecifics, and in particular patterns of BM with walking humans. However, attempts to understand how the brain extracts information from the BM of human walking have so far focused almost exclusively on processing within the *visual* system. A seminal study by Gunnar Johansson (1973) reported that actors defined solely by point lights attached to the major joints of an otherwise invisible body are, once set in motion, highly perceivable. Since then, these so-called point light walkers (hereafter PLWs) have been considered a classic 'form-from-motion' stimulus, and have been deployed as the gold standard tool used to further understand the perceptual and neural mechanisms supporting BM perception. Perception of such PLW patterns survives information degradation, as when PLWs are embedded in 'noise' that mimics the motion of 'signal' point lights [1,2], and when the number of point lights is reduced to represent the major bodily joints only [3,4].

Numerous other PLW studies confirm the remarkable proficiency of human visual BM perception, by showing that observers can quickly discern characteristics including the sex [3,5–10], age [3,8], vulnerability [11], and even emotions of the actor being depicted [12–14]. Further, Loula and colleagues showed that human observers are more sensitive to their own action compared to friends' and strangers' actions during identification and discrimination tasks, suggesting that motor signals play an important role for BM analysis [15]. Another study involving hemiplegic patients with motor system lesions revealed degraded visual sensitivity to point-light actions that correspond to their compromised limbs, but not to point-light actions that correspond to their functional limbs [16]. Furthermore, when asked to estimate the terminal location of a moving point-like arm that vanished after 60% of its movement, observers improved their performances when self-generated movements were presented [17]. All these results highlight the motor contributions to BM perception and support the common coding theory according to which actions are represented through perceptual and motor coding systems [18]. Other results support this theory, showing for instance that producing a running activity briefly prior to the task improved participants' perceptual judgements regarding the direction of a point-light runner [19], that observers' own activities influence the perception of activities of other people [20] or that observers demonstrate maximum sensitivity to actions most familiar to them and reduced sensitivity to actions unfamiliar to them [21] (see [22] and [23] for a review). These findings are consistent with the idea that observers spontaneously simulate, in their own sensorimotor planning system, the actions they observe [24,25], simulation that partially mediates action perception by embodying observed actions [23,26,27].

Furthermore, it has been shown that tactile sensations contribute not only to coding the properties of the external object but also to the formation of a mental body representation, defined as "an abstract representation of one's own body, derived from sensory input but capable of being dissociated from it, and playing a functional role in perception and/or action" [28]. These representations, containing representations of the body itself, are thought to reciprocally influence primary tactile processing and to modulate not only the perception of one's own body, but also the perception of other external object. Specifically, it has been shown that visual information related to the body could affect tactile sensation, a cross-modal effect termed visual enhancement of touch (VET) (see [28] for a review).

Thus, based on the observation of the contribution of motor mechanisms in BM processing, the cross-modal visuo-tactile mechanisms in the perception of bodily stimuli [28], and the finding of highest BM sensitivity to one's own BM stimuli [15], we hypothesized that

perceptions of visually-defined BM would be subject to the influence of somatosensory input. For this, we used a two-interval forced choice paradigm and tested discrimination of PLW speed in the presence of tactile cues. We carried out two experiments aiming at evaluating the role of tactile signals for BM perception along three main lines. First, the discriminability of visual BM was compared when tactile stimulation was applied to the feet (experimental tactile condition—ET) or to the forearms (body-site control tactile condition—CT), compared to a baseline condition without tactile stimulation (only vision condition–V). Assuming that tactile-visual interactions are specific to the foot stimulation, we predicted that task-relevant tactile stimulation (ET) would enhance performance discriminating information from PLWs. Specifically, it was predicted that such stimulation would result in subjects detecting *smaller* differences in speed between walkers across presentation pairs. By contrast, it was predicted that the relatively task-irrelevant tactile forearm stimulation (CT) condition would result in performance outcomes that were comparable to those in the baseline condition (V). Second, we assessed the temporal specificity of somato-visual interactions. Within the audio-visual motion processing literature [29,30], cross-modal integration is optimized under conditions of temporal co-incidence. If the somatosensory-visual BM processing effect is of a similar nature, we expected that temporally coincident (synchronous) foot stimulation (i.e. tactile cue is applied when the seen PLW touches the ground) would result in better performance than asynchronous stimulation (i.e. tactile cue is randomly applied during the PLW cycle). Inclusion of such a manipulation also allowed us to test an alternative account of the effects observed in the first experiment, within which the application of tactile stimulation may have had its effect solely by focusing attention on the body part most relevant to performance on the visual task (see [31] and [4] for a discussion of the significance of the lower limbs for PLW interpretation). In other words, it allowed us to test whether the effects we observed may have arisen via *attention modulation* mechanisms rather than BM-specific cross-modal effects per se. Were that the case then asynchronous and synchronous stimulation of the relevant body part (the foot) should similarly facilitate visual performance. Finally, we tested whether the aforementioned tactile-visual interactions were related to global BM perception or related to other lower-level motion-processing mechanisms. To that end, we relied on the classic method of presenting PLW upside-down, as it is known to specifically impair BM processing while conserving all local motion features as their upright counterparts [4,32,33]. Changing orientation from upright to inverted impede spontaneous recognition of PLWs, making for instance more difficult to detect a camouflaged PLW within a mask [31]. Furthermore a recent study using 9.4T functional MRI has shown that different neural circuits are activated for inverted or upright PLWs processing, with activation of left hemispheric anterior networks engaged in decision making and cognitive control for inverted BM, and activation of right hemispheric multiple networks in response to upright BM [34]. We here quantified the impact of tactile cues on the discriminability of inverted PLW and predicted that performance for upright PLWs would be better than for their inverted counterparts.

## Materials and methods

### Subjects

A total of 43 participants were recruited (22 in Experiment 1, mean age: 27.19 years +/- 3.43 years SD; 8 women, binomial test: p = 0.38; 21 in Experiment 2: mean age: 24.05 years +/- 2.59 years SD; 7 women, binomial test: p = 0.19). Participants in experiments 1 and 2 were not the same. Participants were recruited through printed and electronic advertisements on notice boards at various sites in the Ecole Polytechnique Fédérale de Lausanne (EPFL). After contacting the experimenter, participants received the participant information sheet explaining the

procedure and the goal of the study as well as the exclusion criteria (uncorrected vision deficit, somatosensory deficit). Individuals had normal or corrected-to-normal vision and had no known somatosensory processing deficits. All participants were naive to the purpose of the study.

The local Ethical Committee of the Ecole Polytechnique Fédérale de Lausanne (IRB of EPFL; Switzerland) approved the experiment, which was in accordance with the Declaration of Helsinki (1964). Participants gave written informed consent prior to inclusion in the study and were compensated for their participation (20 Swiss francs).

In Experiment 1, data from one male subject were excluded after that individual interrupted the experiment multiple times. Furthermore, 4 subjects (1 male and 3 female) in Experiment 1 and 3 subjects (2 male and 1 female) in Experiment 2 were excluded from analyses due to attention deficit during the task, i.e. more than half of the catch trials were performed as false alarm (see below procedure part for more details). Thus, it remained 17 subjects (5 women) in Experiment 1 and 18 subjects (6 women) in Experiment 2 for the analysis.

## Apparatus & stimuli

**Visual stimuli.** PLWs were generated using Matlab (Mathworks, Natick, MA) with extensions from the Psychophysics Toolbox [35] and comprised 15 point lights representing the ankles, knees, hips, wrists, elbows, shoulders, center of the pelvis, sternum and head of a walking individual facing the observer. The PLWs were the same for all participants. Individual point lights had a radius of 10 pixels. Presented on a ViewSonic Graphic Series G90 f+ 19 inch monitor (resolution = 1280 x 1024) positioned in front of subjects at a viewing distance of 85 cm, individual PLWs had the size of 50% of the screen's height. Individual point lights were set at 50% contrast with the grey background. The reference frequency of the gait cycle was 0.77 cycle per second. PLW movies consisted of 129 frames representing a full gait cycle of two footsteps, lasting 1298 ms. An interval of 1500 ms separated the presentation of this reference PLW to a second PLW with a slightly faster gait frequency (0.77 + Δf cycle/sec) according to the participants performance (staircase adaptative procedure, described below). Pilot testing confirmed that all variations gave rise to strong perceptions of fluid BM.

**Tactile stimuli.** Somatosensory stimulation was delivered via two matching custom-built devices each comprising twelve 18mm Shaftless Vibration Motors (Precision Microdrives) fitted into small rubber mats (Fig 1). Depending on the condition, each mat was attached either to subjects' feet, or to their forearms. Mats were carefully positioned to ensure that all motors were in contact as confirmed by the subjects. Tactile stimulation to each site was confirmed to be suprathreshold in pilot studies. Individual motors were driven via an LPT-1 port using MatLab such that onset and offset was simultaneous across motors. The duration of the tactile stimuli was adapted depending on the the duration of the visual stimulus (i.e. when PLWs walked faster, tactile stimulations were shorter) and corresponded to 20% of the duration of the total PLW gait.

In the 'synchronous' conditions of Experiment 1 and 2, stimulation onset of the laterally matched body part was timed to coincide with the point at which the PLW 'ankle' dot reached its lowest value on the y-axis. Indeed, for a full phase of PLW cycle, Foot 1 was stimulated between 10–30% of the visual stimuli cycle and Foot 2 between 60–80% of this cycle.

In the 'asynchronous' condition of Experiment 2, tactile stimulation onset was randomized during the visual stimulus. Both frequency and phase were different for the visual stimuli and the tactile stimuli. The duration of the tactile stimulation was the same as in the synchronous condition (corresponding of 20% of the total PLW gait cycle), but for each foot the phase was randomly selected. For example, this could result in Foot 1 being stimulated from 0–20% of

# EXPERIMENT 1

3 conditions:

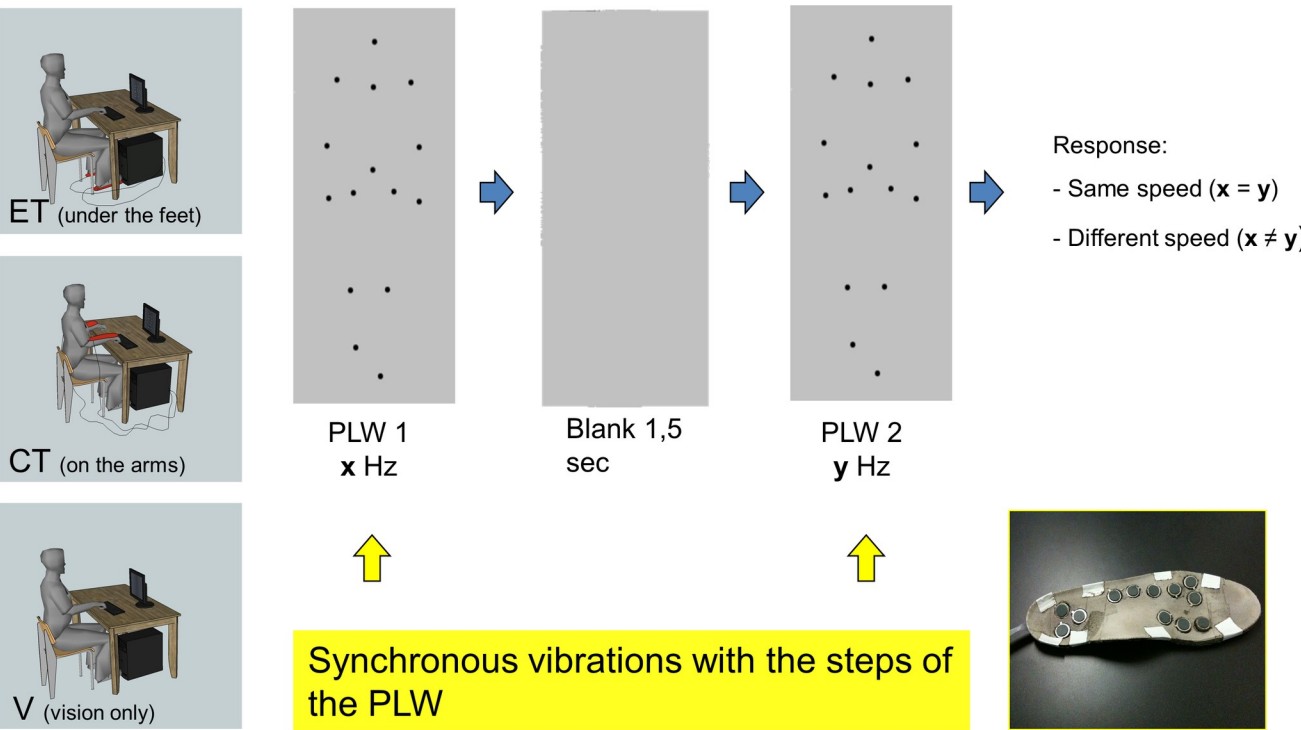

**Fig 1. Experiment 1 procedure.** The subject is sitting in front of a computer screen, where two successive point light walkers (PLW) are shown. Subject has to focus on the visual stimuli and to determine if the PLWs gait speed is the same or different. Depending on the conditions, a vibrating sole (shown in the bottom-right corner) is banding to his feet (ET experimental tactile), his forearms (CT control tactile) or not (V vision only). The vibrations are delivered synchronously with the steps of the PLWs.

the cycle and Foot 2 from 15–35%. The phase was randomized for each foot separately, i. e. it could also be that Foot 1 and Foot 2 were stimulated at the same time.

During the entire experiment, white noise was delivered to the participants through head-phone to avoid any auditory cue related to motors sounds.

## Procedure

Experiments were conducted in an experimental sound insulated and dimmed booth. Subjects were seated directly in front of the monitor with their legs resting comfortably on a box, their feet suspended above the ground, and their wrists (not their forearms) rested upon the table (Fig 1). Such body positioning allowed the tactile stimulation motors to be easily affixed to the targeted body part in each of the relevant conditions.

PLW presentation pairs were separated by 1500 ms intervals. A classical staircase adaptive procedure was used to determine the just noticeable difference between a reference PLW gait speed and a slightly different PLW gait speed. The frequency of the reference stimulus was 0.77 cycle/sec and the adaptive procedure determined an always positive small value $\Delta f$ cycle/sec to add to the second stimulus, which then had a gait frequency of $0.77+\Delta f$ cycle/sec. In each trial, however, we randomly selected with equal probability if the reference stimulus was

shown as stimulus 1 or as stimulus 2, so the second PLW in each trial could walk either faster or slower than the first PLW in that trial.

For each pair, subjects were required to indicate via keypress whether the two seen PLWs had the same or a different gait speed. On that basis, performance for individual participants represents the average difference at which speed discrimination was possible, with lower values indicating higher sensitivity. The experiment consisted of 50 trials per condition. In order to ensure subjects were comfortable with the task, all completed 10 practice trials prior to the start of the actual experiments.

In addition, there was a ¼ chance in each trial that both the first and the second PLW were the reference stimulus with gait frequency of 0.77 cycle/sec (catch trial). The responses to those trials were excluded from the adaptive procedure. These catch trials were introduced to keep the task challenging and to maintain the attention of the subjects. If the correct response ratio for these catch trials was below 50%, meaning that more than half of the responses were false alarm, the participant was excluded from the analyses (4 subjects in Experiment1 and 3 subjects in Experiment 2). This attention deficit is likely due to the length of the experiment (180 repetitions of PLWs pairs in Experiment 1 and 240 repetitions of PLWs pairs in Experiment 2).

## Experiment 1

The first experiment was designed to test whether tactile stimulation would enhance vision-based performance discriminating information from PLWs. Three conditions were ran in a randomized order: the experimental tactile condition (ET) with tactile stimulation to the feet, the control tactile condition (CT) with tactile cues presented to the subjects' forearms and a baseline condition, vision only (V) without tactile stimulation (Fig 1).

### Results

Data were analyzed using a mixed effects probit regression with response as the dependent variable, stimulus intensity (defined as the speed difference between the two PLWs) and condition as fixed effects, with a random intercept by subject and random slopes for each fixed effect. The model revealed a clear effect of stimulus intensity ($X^2$ = 58.40, p < 0.001) indicating that participants performed the task accurately, with a tendency to respond "different" more frequently as speed difference increased, and a main effect of condition ($X^2$ = 7.19, p = 0.03) indicating that our manipulation slightly changed response bias (Fig 2). Importantly, the model revealed an interaction between intensity and condition ($X^2$ = 10.08, p = 0.007), reflecting a steeper slope between response and intensity in the experimental (ET) vs. control tactile condition (CT) with tactile cues applied to the forearms (Fig 2; probit estimate = 4.44, z = 3.08, p = 0.002), but was not found in the visual only (V) vs. control tactile condition (CT) (probit estimate = 0.94, z = 0.67, p = 0.50). This indicates that visuotactile cues delivered to the feet, but not the upper limbs, improves the discrimination of BM perception. In addition, there was a significant difference between the experimental visuotactile and visual condition (probit estimate = 3.07, z = 2.10, p = 0.04), indicating a steeper slope between response and intensity in the ET condition.

## Experiment 2

In order to further explore the parameters of the effect of Experiment 1, we ran in the second experiment a two by two factorial design with one factor as the congruency of the tactile stimulation (Sync and Async conditions) and a second factor as the orientation of the visual stimuli

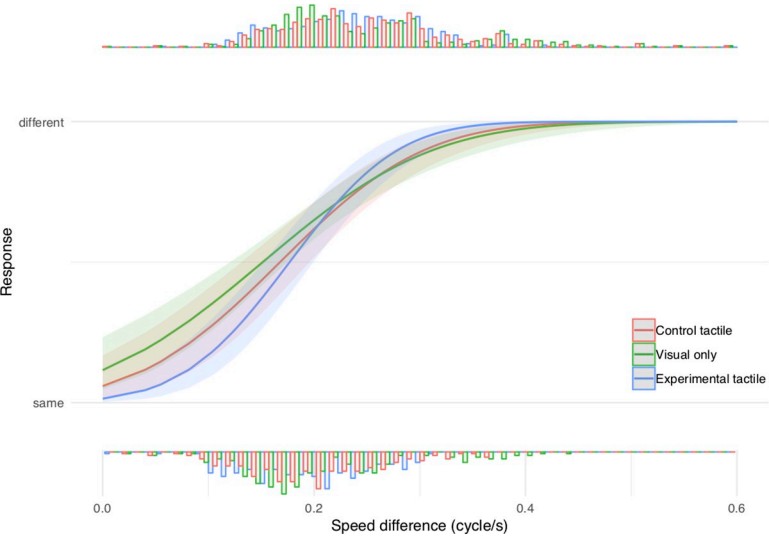

**Fig 2. Experiment 1 results.** Mixed effects probit regression between response and stimulus intensity for the experimental tactile (ET in blue), control tactile (CT in red), and visual condition (V in green). The histograms indicate at the top the distribution of yes (different speed of the PLWs) responses and at the bottom the distribution of no (same speed of the PLWs) responses. Each histogram represents the density of a given response distribution. These results show that additional tactile stimulus had an effect on BM discriminability only if the tactile stimuli a delivered under the feet of the observers.

(Upright and Inverted conditions) (Fig 3). Each of the four conditions were presented in a randomized order.

## Results

Data were analyzed using a mixed probit regression with response as the dependent variable, stimulus intensity (defined as the speed difference between the two PLWs), temporal condition and orientation as fixed effects, with a random intercept by subject and random slopes for intensity and orientation. The model revealed an interaction between orientation and stimulus intensity ($X^2 = 13.48$, $p < 0.001$), revealing better performance in the upright vs. inverted condition. Importantly, a significant interaction between intensity and synchrony ($X^2 = 4.77$, $p = 0.029$) revealed a steeper slope for synchronous vs. asynchronous stimulation, suggesting that BM perception was better in conditions when the tactile foot cue coincided and the PLW 'ankle' dot reached its lowest value at the same time. However, unlike our prediction, no interaction between orientation and synchrony was found ($p = 0.99$), indicating that the effect of synchrony was not specific to upright PLW (Fig 4).

## Discussion

The primary aim of these experiments was to investigate possible effects of somatosensory cues on the perception of visually defined biological motion. We designed Experiment 1 to compare the discriminability of visual BM when tactile stimulation was applied to the participants' feet (ET) or forearms (CT), or when no tactile stimulation was applied (V). Experiment 2 was aimed to further assess the temporal specificity of somatosensory-visual interactions uncovered in Experiment 1, and to test whether they were specific to BM processing.

Considered collectively data from the conditions tested in Experiment 1 suggest two key findings. Firstly, as we found differences in BM discriminability between the experimental condition (ET) and the visual only (V) condition, they indicate that the additional tactile

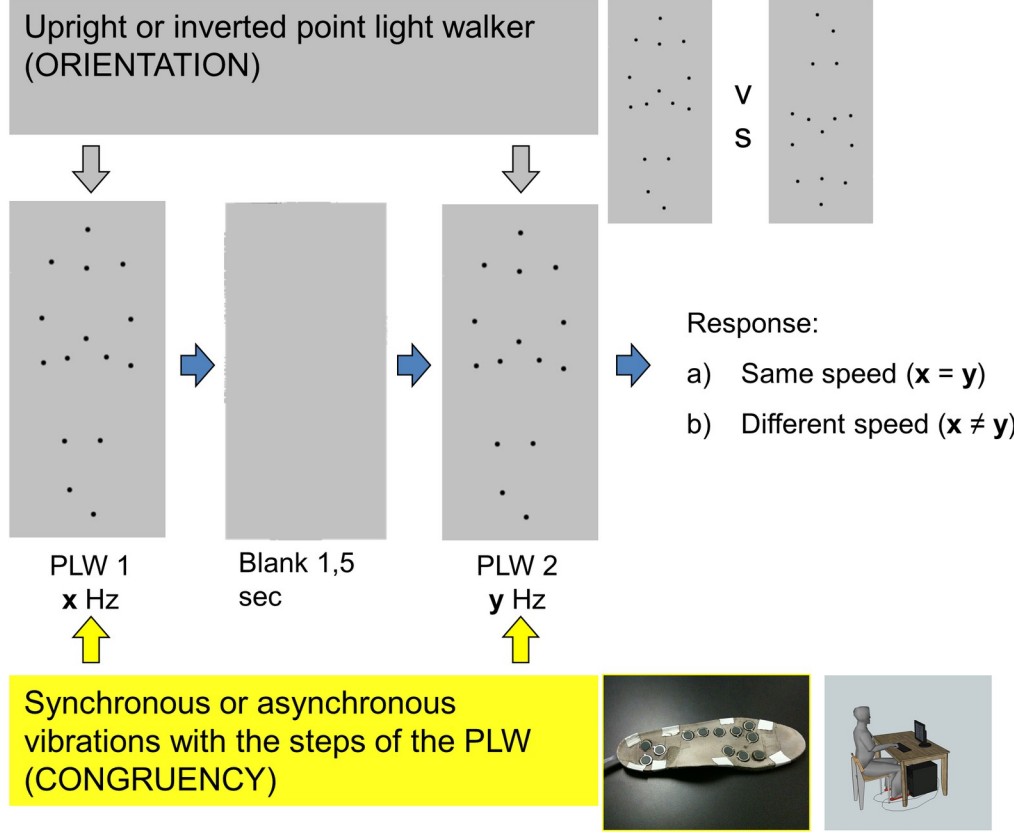

**Fig 3. Experiment 2 procedure.** The subject is sitting in front of a computer screen, where two successive point light walkers (PLW) are shown. Subject has to focus on the visual stimulus in order to determine if the PLWs gait speed is the same or different. A vibrating sole is banding to his feet. The specificity of the effect on BM perception is tested by presenting an upright PLW (Upright condition) or an inverse PLW (Inverted condition). In order to investigate the cross-modal integration of the tactile and visual stimuli, their temporal coincidence is modulated in a way that the vibrations are either synchronous (Sync condition) or asynchronous (Async condition) with the steps of the PLWs.

stimulus per se had an effect on BM perception. Secondly, the tactile effect on vision-based discrimination was body-part specific, with facilitation being observed when tactile stimulation was delivered to subjects' feet (ET), but not when the same stimulus was applied to a non-relevant (forearm) body part (CT). As the frequency and the phase of both visual and tactile stimuli were the same in this experiment, the tactile input itself (through its frequency and its duration) could have conveyed additional information about the speed of the PLW. It is then possible that the improvement in BM discriminability was only due to this additional information. However, the finding that the same tactile stimuli improved BM discrimination only when they were delivered under the feet makes this explanation unlikely. There was indeed no difference in BM discrimination when no tactile stimulation occurred (V) and when tactile stimulation was applied on the forearms (CT). Furthermore, given that tactile stimulation to each site was qualitatively matched in pilot studies (and informally by each subject during the experimental set-up), it is unlikely these effects arise through any dissimilarity in tactile perception across stimulation sites, although we cannot exclude it.

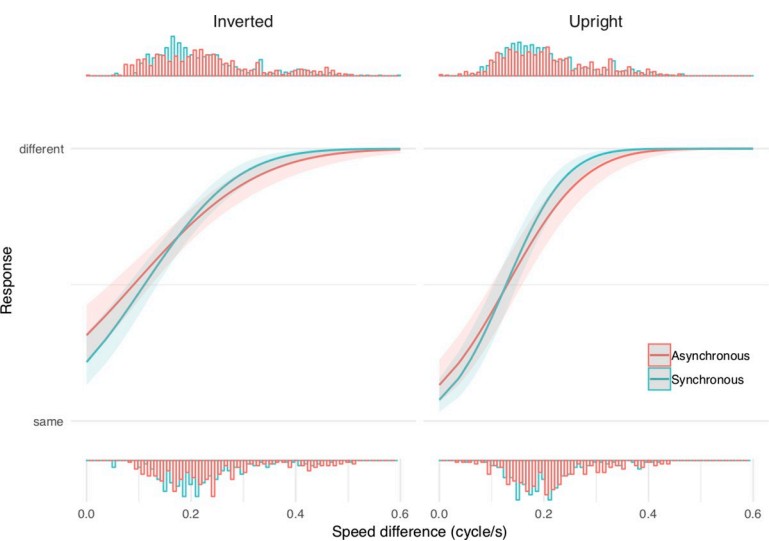

**Fig 4. Experiment 2 results.** Mixed effects probit regression between response and stimulus intensity as a function of temporal condition (synchronous (blue) or asynchronous (red) tactile stimulation) for inverted orientation (left panel) and upright orientation (right panel). Each histogram represents the density of a given response distribution. These results show that tactile stimuli synchronously timed with the visual stimuli systematically improved BM discriminability. Both BM and non-BM (inverted visual stimulus) were better discriminated in the presence of synchronous tactile cues.

The manipulation of congruency in Experiment 2 shows that improvement in BM discriminability with feet tactile stimulation only happened when the tactile stimulation was delivered synchronously with the visual stimulus. This indicates that the increase in discriminability was driven by the coherent interaction between the visual and the tactile stimuli (cross-modal interaction). This finding fits well with the *temporal rule* of the multisensory integration theory [36], stating that cross-modal integration occurs when the constituent unisensory stimuli arise at approximately the same time. Furthermore, as predicted, results of Experiment 2 argue against a purely attention-modulated account: if tactile foot stimulation simply directed visual attention towards the body part most relevant for making the visual discrimination, then performance in the asynchronous condition should have yielded comparable performance levels to those in the synchronous condition. This was not the case. However, as the frequency and the phase are different for the visual and tactile stimuli in the asynchronous condition, we cannot exclude, based only on Experiment 2, that the improvement in BM discriminability in the synchronous condition comes from additional information about the speed of the PLW conveyed by the tactile stimulation itself.

The manipulation of the orientation of the PLWs showed that this effect of synchronous visuotactile stimulation occurred equally for upright and inverted PLW. Orientation manipulation of PLW is known to have a catastrophic effect on ability to extract information from PLWs: as in the case of static face representations, stimulus processing is adversely impacted when figures are presented upside down [4,12,32], even if observers are informed of seeing an inverted visual stimulus [32], a finding that has been taken as evidence that the neural mechanisms subserving BM processing are orientation-tuned. Although we found that discriminability was overall better with upright visual stimuli, our results indicate that both BM and non-BM were better discriminated in the presence of synchronous tactile cues. This finding suggests that synchronous tactile cues does not interact in the extraction of BM features.

Taken together, the present findings extend the demonstration of cross-modal effects on motion processing to visuotactile cues, in line with what has been shown in the audiovisual domain [37,38].

However, this study presents several limitations. First, our sample of participants was not balanced for gender. Only 8 women of 22 subjects participated in Experiment 1 and 7 women of 21 subjects in Experiment 2. Previous studies have shown differences in BM perception depending on gender, showing for instance that female and male observers were found to judge differently whether a frontal PLW is facing toward or backward them [39], as well as emotional expressions [40]. Of note, post-hoc analysis revealed better performance in PLW discrimination in male observers in Experiment 2 (77.1% vs. 74.4% t(10.05) = 2.48, p = 0.03), but not in Experiment 1 (t(12.19) = 1.41, p = 0.18). A supplementary binomial mixed-effects regression including gender as a covariate in Experiment 2 revealed that the interaction between intensity and synchrony remained significant ($X^2$ = 11.41, p < 0.001), and was not modulated by gender ($X^2$ = 0.66, p = 0.41). Future studies will be required to assess whether this effect is anecdotal or whether cross-modal effects on motion processing depends on gender. Secondly, we cannot exclude the possibility that differences in tactile perception between the forearms and the feet may have resulted from difference in sensitiveness between these two body parts. We hypothesized during the study conception that the feet would be more relevant than forearms regarding walking actions, and therefore predicted that performance would be better with tactile stimulation on this site than elsewhere on the body. Finally, we also note that our stimulus combination may not be an ecologically valid representation of external walking cues (it is unlikely that a seen walker also generates reliable externally generated somatosensory cues, even at a subthreshold level). As we observed an effect only when the tactile stimulation was applied under the feet in Experiment 1, we speculate this effect could arise by tapping into the very system within which external/internal neural action representations merge. In other words, it is possible that externally sourced somatosensory stimulation may up-regulate the usual vision-based embodiment of the observed walking action (for established cases of somatosensory-visually driven stimulus embodiment see for e.g. [41]; [42]). This hypothesis goes in line with the idea that tactile sensation contribute to a body representation, which in turn influences the perception of external object [28]. Loula and colleagues' results, having highlighted the motor experience contribution to BM perception, suggest also that cues on the observer could influence perception of others, as a self-influence on BM perception [15]. In line with this role of motor simulation in BM perception, it is possible that tactile stimulations under the feet (and not on the forearms) improve the observers' motor simulation of the walking motion in their sensorimotor planning. This simulation, so reinforced by tactile congruent cues, could then improves the discriminability of PLW. Another study report such possible reinforcement of body representation by multimodal perception: investigating agency for walking avatars, Menzer and colleagues (2010) has shown that agency for one's own footsteps decreased rapidly when a temporal delay was introduced between the participants' footsteps and the auditory consequences of those footsteps [43]. Interestingly the results were comparable with those of studies manipulating visual cues [44,45], reflecting thus common, supramodal, mechanisms in the conscious action monitoring of auditory and visual action consequences [43], mechanisms that are compatible with the aforementioned forward sensory system, that is independent of the sensory modality tested. How precisely such embodiment influences perceptual sensitivity to the motion of 'another' is the subject for future research.

## Acknowledgments

We gratefully thank all participants of this study, as well as the three anonymous reviewers for their helpful comments.

## Author Contributions

**Conceptualization:** Pierre Progin, Manuel Mercier, Lars Schwabe, Olaf Blanke.

**Data curation:** Pierre Progin, Anna Brooks, Lars Schwabe.

**Formal analysis:** Pierre Progin, Nathan Faivre, Wenwen Chang, Manuel Mercier, Lars Schwabe.

**Funding acquisition:** Wenwen Chang.

**Methodology:** Pierre Progin, Manuel Mercier, Lars Schwabe.

**Project administration:** Olaf Blanke.

**Resources:** Olaf Blanke.

**Software:** Wenwen Chang, Lars Schwabe.

**Supervision:** Nathan Faivre, Anna Brooks, Kim Q. Do, Olaf Blanke.

**Validation:** Nathan Faivre, Kim Q. Do, Olaf Blanke.

**Visualization:** Pierre Progin.

**Writing – original draft:** Pierre Progin, Anna Brooks.

**Writing – review & editing:** Pierre Progin, Nathan Faivre, Anna Brooks, Olaf Blanke.

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
