## [Decision Letter · Decision Letter 0]

3 Sep 2019

PONE-D-19-17841

Somatosensory-visual effects in visual biological motion perception

PLOS ONE

Dear Dr. Progin:

Thank you for submitting your manuscript to PLOS ONE. After careful consideration, we feel that it has merit but does not fully meet PLOS ONE’s publication criteria as it currently stands. Therefore, we invite you to submit a revised version of the manuscript that addresses the points raised during the review process.

 Two experts in the field had a look at your submission, and both of them find your work timely and interesting. However, both of them have several suggestion how to further improve your manuscipt. Please address all comments on the point-by-point basis in your rebuttal letter to me.

We would appreciate receiving your revised manuscript by Oct 18 2019 11:59PM. To enhance the reproducibility of your results, we recommend that if applicable you deposit your laboratory protocols in protocols.io, where a protocol can be assigned its own identifier (DOI) such that it can be cited independently in the future. For instructions see: http://journals.plos.org/plosone/s/submission-guidelines#loc-laboratory-protocols

We look forward to receiving your revised manuscript.

Kind regards,

Marina A. Pavlova, PhD

Academic Editor

PLOS ONE

Journal Requirements:

1. We note that you have stated that you will provide repository information for your data at acceptance. Should your manuscript be accepted for publication, we will hold it until you provide the relevant accession numbers or DOIs necessary to access your data. If you wish to make changes to your Data Availability statement, please describe these changes in your cover letter and we will update your Data Availability statement to reflect the information you provide.

Reviewers' comments:

Reviewer's Responses to Questions

**Comments to the Author**

1. Is the manuscript technically sound, and do the data support the conclusions?

Reviewer #1: Yes

Reviewer #2: Yes

2. Has the statistical analysis been performed appropriately and rigorously? 

Reviewer #1: Yes

Reviewer #2: Yes

3. Have the authors made all data underlying the findings in their manuscript fully available?

Reviewer #1: Yes

Reviewer #2: No

4. Is the manuscript presented in an intelligible fashion and written in standard English?

Reviewer #1: Yes

Reviewer #2: Yes

5. Review Comments to the Author

Reviewer #1: Authors depeen somatosensory effect, in particular tactile foot stimulation, on BM visual-perception, basing on evidence of brain areas integrating auditory and visual signals. I found the study interesting and the purpose of the research relevant. I would suggest some clarifications in order to make the manuscript clearer:

ABSTRACT: Even though it explains and justifies the aim of the research, it does not give space to the method. I suggest adding method of the research in term of the design, number of experiments, tasks, participants....

INTRODUCTION: I found the introduction section well written, reporting background related to research questions. Nevertheless, I did not find a section showing literature that lead authors to investigate temporal specificity of somato-visual interactions. I would suggest integrating in introduction this part.

When authors present experiment 1 conditions (ET, CT,V), while they well describe ET and CT, they do not specify baseline condition (V).

METHODS: I would suggest including chi-square analysis to report differences in gender distribution (females:males) in both experiments. In experiment 2 7/21 females are included, that means they are 1/3 of the sample. Is there a significant difference between males and females? A growing literature is documenting a gender effect in BM perception (Sokolov et al.,Frontiers in Psychology, 2011; Krüger et al., 2013, PloS One; Pavlova et al., 2014, Cerebral Cortex...).

I also suggest adding more information regarding recruitment of participants: How and where participants were recruited? What about their education level? Which were inclusion and exclusion criteria?

Are PLW stimuli balanced in males and females stimuli?

DISCUSSION: I would suggest starting discussion with a resume of the research question and experiment design.

Reviewer #2: This paper aims to examine the somatosensory effects in visual biological motion perception. For this, the authors propose two studies where they used a two-interval forced choice paradigm and tested the discrimination of point-lights walkers (PLW) speed in the presence of tactile cues. More precisely, the first experiment compares the performances in speed discrimination of PLW when the visual presentation is associated or not or incongruent tactile cues. The second experiment tests the role of time synchrony and recognition of the global motion in the tactile effects. Globally, the analyses show that the speed discrimination of PLW is improved when the visual stimulation is associated with congruent tactile cue (stimulation on the feet) but this effect is only observed when the tactile stimulation is synchronous with the visual stimuli. Interestingly, this effect does not rely with the global recognition of PLW because it persists even the PLW are presented upside-down. The authors interpret their results as somatosensory-visual interactions in BM perception with particular reference to the human mirror neuron system and multisensory mechanisms in action perception.

Globally, I find the study timely and very interesting. Introduction is well documented, and methods and results are clear. However, I have some suggestions to improve the manuscript.

Introduction:

I find the introduction clear and well documented even the authors could cite more recent studies about PLD.

See for example

- Bidet-Ildei, Chavin & Coello, 2010 for the discriminability on PLD in scrambled masking dots

- Martel, Bidet-Ildei & Coello, 2011 for the sensitivity to the own actions.

I think that it misses in introduction something about the upside-down effect. It is surprising since this effect is specifically studied in Experiment 2.

Method:

-Are the same participants in Experiment 1 and 2?

- Why the authors choose to describe both experiments and after to present the results? I find difficult to follow. I think that the manuscript could be clarified if the authors finished first the Experiment 1 and after proposing the Experiment 2. Moreover, in this configuration, they can more specified the specific objectives of each experiment and to add a specific discussion.

Discussion

The authors explain their results with sensorial interactions but is it possible to imagine that their results could be due to more precise motor simulation of the PLW?. Indeed, it would be logical that a tactile stimulation on the feet can reinforce the motor simulation of a walking motion and so improve the discriminability of PLW? Even this framework has difficulty to explain the results in Experiment 2 (the tactile effect with upside-down PLW), I think important to envisage this possibility. If the authors cut their study in two separate experiments, this hypothesis could be envisaged in the discussion of Experiment 1.

6. PLOS authors have the option to publish the peer review history of their article (what does this mean?). If published, this will include your full peer review and any attached files.

Reviewer #1: No

Reviewer #2: Yes: Christel Bidet-Ildei

---

## [Author Response · Author response to Decision Letter 0]

7 Oct 2019

RESPONSE TO REVIEWERS

Dear Dr. Pavlova,

Thank you for giving us the opportunity to submit a revised draft of our manuscript titled “Somatosensory-visual effects in visual biological motion perception” to PlosOne. We appreciate the time and effort that you and the reviewers have dedicated to providing your valuable feedback on our manuscript. We are grateful to the reviewers for their insightful comments on our paper. We have been able to incorporate changes to reflect most of the suggestions provided by the reviewers. We have highlighted the changes within the manuscript.

Here is a point-by-point response to the reviewers’ comments and concerns. Please, find below the reviewers’ comments repeated in italics and our responses inserted after each comment.

We look forward to hearing from you in due time regarding our submission and to respond to any further questions and comments you may have.

Sincerely,

 Pierre Progin and co-authors

Comments from Reviewer #1: Authors depeen somatosensory effect, in particular tactile foot stimulation, on BM visual-perception, basing on evidence of brain areas integrating auditory and visual signals. I found the study interesting and the purpose of the research relevant. I would suggest some clarifications in order to make the manuscript clearer:

ABSTRACT: 

Even though it explains and justifies the aim of the research, it does not give space to the method. I suggest adding method of the research in term of the design, number of experiments, tasks, participants....

Response: Thank you for pointing this out. We agree with this comment. Therefore, we have added information about the methodological part of the study in the abstract:

In two experiments, we asked healthy participants to perform a speed discrimination task on two point light walkers (PLW) presented one after the other. In the first experiment, we quantified somatosensory-visual interactions by presenting PLW together with tactile stimuli either on the participants’ forearms or feet soles. In the second experiment, we assessed the specificity of these interactions by presenting tactile stimuli either synchronously or asynchronously with upright or inverted PLW.

INTRODUCTION: 

I found the introduction section well written, reporting background related to research questions. Nevertheless, I did not find a section showing literature that lead authors to investigate temporal specificity of somato-visual interactions. I would suggest integrating in introduction this part.

Response: We agree with this point and have revised the introduction accordingly. A first reference to a seminal neuroanatomical study (Meredith 1987) was added in the contextual part of the introduction:

Several studies have now shown that auditory information influences visual BM processing across a range of tasks (10,46–48), given that the two signals are temporally linked (49). As Meredith and Stein found in cat superior colliculus (50), multimodal neurons in STS are more responsive to coincident and colocalized multimodal stimuli compared to spatially and temporally incoherent stimuli (51). 

A reference to audio-visual integration was also added to the description of the second experiment at the end of the introduction:

Within the audio-visual motion processing literature (49,51), cross-modal integration is optimized under conditions of temporal co-incidence. If the somato-visual BM processing effect is of a similar nature, we expected that temporally coincident (synchronous) foot stimulation (i.e. tactile cue is applied when the seen PLW touches the ground) would result in better performance than asynchronous stimulation (i.e. tactile cue is applied at other times). 

When authors present experiment 1 conditions (ET, CT,V), while they well describe ET and CT, they do not specify baseline condition (V).

Response: We would like to thank the reviewer for noticing this omission. We now describe the baseline condition in the introduction:

First, the discriminability of visual BM was compared when tactile stimulation was applied to the feet (experimental tactile condition - ET) or to the forearms (body-site control tactile condition - CT), compared to a baseline condition in which no tactile stimulation was applied (only vision condition – V).

METHODS: 

I would suggest including chi-square analysis to report differences in gender distribution (females:males) in both experiments. In experiment 2 7/21 females are included, that means they are 1/3 of the sample. Is there a significant difference between males and females? A growing literature is documenting a gender effect in BM perception (Sokolov et al.,Frontiers in Psychology, 2011; Krüger et al., 2013, PloS One; Pavlova et al., 2014, Cerebral Cortex...).

Response: We agree with this and have run a binomial test to know if our groups were significantly unbalanced for gender. In Experiment 1, we had 8 females on 21 participants, which corresponded to a p-value = 0.38. In Experiment 2, we had 7 females on 21 participants, with a corresponding p-value = 0.19. Given that groups were not significantly unbalanced, and considering that our sample size is rather small, we decided not to add this factor as a covariate of interest. However, following the reviewer’s suggestion, we now discuss the relevance of gender for BM perception:

Although our samples were not significantly unbalanced (see methods), the influence of gender on BM perception is worth considering. Indeed, female and male observers were found to judge differently whether a frontal PLW is facing toward or backward them (57), as well as emotional expressions (58, 59). Future studies will be required to assess whether cross-modal effects on motion processing depend on gender. 

I also suggest adding more information regarding recruitment of participants: 

Response: We now mention more details about recruitment:

Participants were recruited through printed and electronic advertisements on notice boards at various sites in the Ecole Polytechnique Fédérale de Lausanne (EPFL). They were all student at the EPFL. After contacting the experimenter, participants received the participant information sheet explaining the procedure and the goal of the study as well as the exclusion criteria (uncorrected vision deficit, somatosensory deficit). 

How and where participants were recruited? 

Response: Participants were recruited through advertisement on notice board in the campus EPFL at Lausanne. 

What about their education level? 

Response: All participants were student at EPFL, in bachelor or master degrees.

Which were inclusion and exclusion criteria?

Response: Inclusion criteria were:

1. Age between 18 and 60 years.

2. Willing to refrain from drinking alcohol at the testing days and from consuming

psychoactive substances 2 weeks before testing days and for the duration of the

study.

3. Willing and capable to give informed consent for the participation in the study

after it has been thoroughly explained.

4. Absence of neurological and major physical impairment, with normal or corrected to normal sensory abilities.

5. Informed consent form was signed.

Exclusion criteria were:

1. Prior participation in a similar study that could bias the current results

2. Presence of major internal or neurological disorders.

3. Non-compliance to the instructions of the experimenter or an inappropriate behavior hindering the normal progress of the experiment.

Are PLW stimuli balanced in males and females stimuli?

Response: PLWs were neutral regarding gender features and all stimuli were the same (PLWs shown in a frontal view) for every participant. We now specify this point in the revised manuscript in the “Apparatus and stimuli” part:

The PLWs were the same for all participants.

DISCUSSION: 

I would suggest starting discussion with a resume of the research question and experiment design.

Response: We agree with this suggestion and have modified the discussion accordingly:

The primary aim of these experiments was to investigate possible effects of somatosensory cues on the perception of visually defined biological motion. We designed Experiment 1 to compare the discriminability of visual BM when tactile stimulation was applied to the participants’ feet (ET) or forearms (CT), or when no tactile stimulation was applied (V). Experiment 2 was aimed to further assess the temporal specificity of somatosensory-visual interactions uncovered in Experiment 1, and to test whether they were specific to BM processing.

Comments from Reviewer #2: This paper aims to examine the somatosensory effects in visual biological motion perception. For this, the authors propose two studies where they used a two-interval forced choice paradigm and tested the discrimination of point-lights walkers (PLW) speed in the presence of tactile cues. More precisely, the first experiment compares the performances in speed discrimination of PLW when the visual presentation is associated or not or incongruent tactile cues. The second experiment tests the role of time synchrony and recognition of the global motion in the tactile effects. Globally, the analyses show that the speed discrimination of PLW is improved when the visual stimulation is associated with congruent tactile cue (stimulation on the feet) but this effect is only observed when the tactile stimulation is synchronous with the visual stimuli. Interestingly, this effect does not rely with the global recognition of PLW because it persists even the PLW are presented upside-down. The authors interpret their results as somatosensory-visual interactions in BM perception with particular reference to the human mirror neuron system and multisensory mechanisms in action perception.

Globally, I find the study timely and very interesting. Introduction is well documented, and methods and results are clear. However, I have some suggestions to improve the manuscript.

Response: We would like to thank the reviewer for noting the quality of our work. 

INTRODUCTION: 

I find the introduction clear and well documented even the authors could cite more recent studies about PLD.

See for example

- Bidet-Ildei, Chavin & Coello, 2010 for the discriminability on PLD in scrambled masking dots

- Martel, Bidet-Ildei & Coello, 2011 for the sensitivity to the own actions.

Response: In line with the reviewer’s suggestion, we now cite these relevant studies in the introduction:

Furthermore, when asked to estimate the terminal location of a moving point-like arm that vanished after 60% of its movement, observers improved their performances when self-generated movements were presented (17).

Other results support this theory, showing for instance that producing a running activity briefly prior to the task improved participants’ perceptual judgements regarding the direction of a point-light runner (19),

I think that it misses in introduction something about the upside-down effect. It is surprising since this effect is specifically studied in Experiment 2.

Response: We fully agree with the reviewer, and apologize for not citing this work earlier. We added a paragraph at the end of the introduction to emphasize this point:

To that end, we relied on the classic method of presenting PLW upside-down, as it is known to specifically impair BM processing while conserving all local motion features as their upright counterparts (4,54,55). Changing orientation from upright to inverted impede spontaneous recognition of PLWs, making for instance more difficult to detect a camouflaged PLW within a mask (54). 

This effect is then mention in the discussion:

Orientation manipulation of PLW is known to have a catastrophic effect on ability to extract information from PLWs: as in the case of static face representations, stimulus processing is adversely impacted when figures are presented upside down (4,12,54), even if observers are informed of seeing an inverted visual stimulus (54), a finding that has been taken as evidence that the neural mechanisms subserving BM processing are orientation-tuned.

METHODS: 

-Are the same participants in Experiment 1 and 2?

Response: No, participants from Experiment 1 and 2 were not the same. We now mention this important point in the method part:

Participants in experiments 1 and 2 were not the same.

- Why the authors choose to describe both experiments and after to present the results? I find difficult to follow. I think that the manuscript could be clarified if the authors finished first the Experiment 1 and after proposing the Experiment 2. Moreover, in this configuration, they can more specified the specific objectives of each experiment and to add a specific discussion.

Response: We would like to thank the reviewer for this advice. The reviewer is right to point that describing both experiment before the results made the reading of the manuscript more difficult. We have now moved the result section of Experiment 1 right after the description of the corresponding methods, and only then describe Experiment 2 and its results. However, we have kept the discussion of both experiments in the general discussion as we think it helps to keep in mind an overview of both experiments before hypothesizing possible mechanisms underlying the reported effects. 

Please note that figure number changed with this new configuration: former Fig. 3 is now Fig. 2, and former Fig. 2 is now Fig. 3.

DISCUSSION: 

The authors explain their results with sensorial interactions but is it possible to imagine that their results could be due to more precise motor simulation of the PLW? Indeed, it would be logical that a tactile stimulation on the feet can reinforce the motor simulation of a walking motion and so improve the discriminability of PLW? 

Response: We fully agree with this point. We consider that tactile stimulation may up-regulate the vision-based embodiment of the observed walking action. Accordingly, it is possible that tactile stimulation improves the simulation of action in subjects’ own sensorimotor planning system. We apologize that this idea was not highlighted more clearly in the manuscript. We now emphasize this hypothesis in the text:

In line with this role of motor simulation in BM perception, it is possible that tactile stimulations under the feet improve the observers’ motor simulation of the walking motion in their sensorimotor planning. This simulation, so reinforced by tactile congruent cues, could then improves the discriminability of PLW. Another study reports such possible reinforcement of body representation by multimodal perception: 

Even this framework has difficulty to explain the results in Experiment 2 (the tactile effect with upside-down PLW), I think important to envisage this possibility. If the authors cut their study in two separate experiments, this hypothesis could be envisaged in the discussion of Experiment 1

Response: The reviewer is right to point that results of Experiment 2 show that the effect isn’t specific to biological motion. This finding of Experiment 2 suggests indeed that synchronous tactile cues does not interact in the extraction of BM features, but in other motion features. We acknowledge that important point in the discussion. 

Although we understand the reviewer’s point of view regarding the discussion format, we preferred discussing both experiments together to provide a more global view on the involved mechanisms. We are willing to change format should the reviewer and editor argue in this sense.

---

## [Decision Letter · Decision Letter 1]

30 Oct 2019

PONE-D-19-17841R1

Somatosensory-visual effects in visual biological motion perception

PLOS ONE

Dear Dr Progin,

Thank you for submitting your manuscript to PLOS ONE. After careful consideration, we feel that it has merit but does not fully meet PLOS ONE’s publication criteria as it currently stands. Therefore, we invite you to submit a revised version of the manuscript that addresses the points raised during the review process.

Two Reviewers who reviewed the previous version of your manuscript submitted their reports. I also had an attentive look at this version. I believe that you have address the following issues: (i) Intro: You refer to work that supports co9nnection between perception and production of body motion. However, there are also reports that are not in aggrement with this view (e.g., from my own lab Pavlova et al., 2003, Brain). There are also several papers (e.g., Pavlova et al., 2017 Cerebral Cortex) with 9.4 fMRI and biological motion that you may wish to discuss.(ii) REviewers did already draw your attention to unbalanced number of females and males in your experiments. Although you replied that binormial analysis does not show significant difference in number of female/male observers, you have to show that there were no gender difference in performance on your tasks to consider these groups homogeneous. (iii) VERY IMPORTANT: In Method section (Subjects) you are writing that several participants (4 in Exp. 1 and 3 in Exp. 2) had been excluded because of attentional problems. You wrote: (see below for more details'. Please explain the reasons in the text where you did mention exclusion, and indicate how many females and males had been excluded, and how many of them  entered your data analysis. (iv) You are writing that your participants were students. However, you did mention that one of your inclusion criteria  was age till 60 yrs. Is it not confusing? Please carefully address these issues in your revision.

We would appreciate receiving your revised manuscript by Dec 14 2019 11:59PM. To enhance the reproducibility of your results, we recommend that if applicable you deposit your laboratory protocols in protocols.io, where a protocol can be assigned its own identifier (DOI) such that it can be cited independently in the future. For instructions see: http://journals.plos.org/plosone/s/submission-guidelines#loc-laboratory-protocols

We look forward to receiving your revised manuscript.

Kind regards,

Marina A. Pavlova, PhD

Academic Editor

PLOS ONE

Reviewers' comments:

Reviewer's Responses to Questions

**Comments to the Author**

1. If the authors have adequately addressed your comments raised in a previous round of review and you feel that this manuscript is now acceptable for publication, you may indicate that here to bypass the “Comments to the Author” section, enter your conflict of interest statement in the “Confidential to Editor” section, and submit your "Accept" recommendation.

Reviewer #1: All comments have been addressed

Reviewer #2: All comments have been addressed

2. Is the manuscript technically sound, and do the data support the conclusions?

Reviewer #1: Yes

Reviewer #2: Yes

3. Has the statistical analysis been performed appropriately and rigorously? 

Reviewer #1: Yes

Reviewer #2: Yes

4. Have the authors made all data underlying the findings in their manuscript fully available?

Reviewer #1: Yes

Reviewer #2: (No Response)

5. Is the manuscript presented in an intelligible fashion and written in standard English?

Reviewer #1: Yes

Reviewer #2: (No Response)

6. Review Comments to the Author

Reviewer #1: I thank authors that fully addressed all my concerns. I find the manuscript suitable for publication.

Reviewer #2: The authors did a very good job of editing. The paper is improved and I highly recommend the publication.

7. PLOS authors have the option to publish the peer review history of their article (what does this mean?). If published, this will include your full peer review and any attached files.

Reviewer #1: No

Reviewer #2: Yes: Christel Bidet-Ildei

---

## [Author Response · Author response to Decision Letter 1]

26 Nov 2019

RESPONSE TO EDITOR

Dear Dr. Pavlova,

Thank you for providing us your valuable feedback on our manuscript titled “Somatosensory-visual effects in visual biological motion perception”. We are grateful for your insightful comments on our paper. We have been able to incorporate changes to reflect most of the suggestions you provided. We have highlighted these changes within the manuscript.

Here is a point-by-point response to your comments and concerns. Please, find below your comments repeated in italics and our responses inserted after each comment.

We look forward to hearing from you in due time regarding our submission and to respond to any further questions and comments you may have.

Sincerely,

 Pierre Progin and co-authors

EDITOR’S COMMENTS TO AUTHOR:

Comments to the Author: Two Reviewers who reviewed the previous version of your manuscript submitted their reports. I also had an attentive look at this version. I believe that you have address the following issues: 

(i) Intro: You refer to work that supports co9nnection between perception and production of body motion. However, there are also reports that are not in aggrement with this view (e.g., from my own lab Pavlova et al., 2003, Brain). There are also several papers (e.g., Pavlova et al., 2017 Cerebral Cortex) with 9.4 fMRI and biological motion that you may wish to discuss.

Response: We thank the Editor for pointing us to these relevant studies, which we now cite in the introduction:

“The planning system might be constitutional for the brain, as motor experience per se doesn’t seem to be necessary for BM perception. A study with young patients with periventricular leukomalacia revealed indeed that patients with early-life impaired motor ability showed the same sensitivity to BM that patients without motor disorder, leading to the assumption that a hard-wired network for both perception and production of BM might be inherent for the brain (28).”

“Furthermore a recent study using 9.4T functional MRI has shown that different neural circuits are activated for inverted or upright PLWs processing, with activation of left hemispheric anterior networks engaged in decision making and cognitive control for inverted BM, and activation of right hemispheric multiple networks in response to upright BM as compared with scarce activation to inverted displays (57).”

(ii) REviewers did already draw your attention to unbalanced number of females and males in your experiments. Although you replied that binormial analysis does not show significant difference in number of female/male observers, you have to show that there were no gender difference in performance on your tasks to consider these groups homogeneous. 

Response: As requested, we computed the average performance across female and male observers. We found no difference for Experiment 1 (t(12.19) = 1.41, p = 0.18) and slightly better performance for males in Experiment 2 (77.1% vs. 74.4% t(10.05) = 2.48, p = 0.03). We now mention this possible issue in the discussion:

“Although our samples were not significantly unbalanced (see methods), the influence of gender on BM perception is worth considering. Indeed, we found no difference for Experiment 1 (t(12.19) = 1.41, p = 0.18) but slightly better performance for males in Experiment 2 (77.1% vs. 74.4% t(10.05) = 2.48, p = 0.03). This is in line with previous findings showing that female and male observers were found to judge differently whether a frontal PLW is facing toward or backward them (59), as well as emotional expressions (60). Future studies will be required to assess whether cross-modal effects on motion processing depend on gender. “

(iii) VERY IMPORTANT: In Method section (Subjects) you are writing that several participants (4 in Exp. 1 and 3 in Exp. 2) had been excluded because of attentional problems. You wrote: (see below for more details'. Please explain the reasons in the text where you did mention exclusion, and indicate how many females and males had been excluded, and how many of them entered your data analysis. 

Response: We apologize if exclusion criteria were not explained in sufficient details. They are specified under the procedure section:

“In addition, there was a ¼ chance in each trial that both the first and the second PLW were the reference stimulus with gait frequency of 0.77 cycle/sec (catch trial). The responses to those trials were excluded from the adaptive procedure. These catch trials were introduced to keep the task challenging and to maintain the attention of the subjects. If the correct response ratio for these catch trials was below 50%, meaning that more than half of the responses were false alarm, the participant was excluded from the analyses (4 subjects in Experiment1 and 3 subjects in Experiment 2).”

We now mention in the subject section that explanation are in the procedure part of the manuscript, and indicate the gender of the 7 excluded subjects:

“In Experiment 1, data from one male subject were excluded after that individual interrupted the experiment multiple times. Furthermore, 4 subjects (1 male and 3 female) in Experiment 1 and 3 subjects (2 male and 1 female) in Experiment 2 were excluded from analyses due to attention deficit during the task, i.e. more than half of the catch trials were performed as false alarm (see below procedure part for more details). Thus it remained 17 subjects (5 women) in Experiment 1 and 18 subjects (6 women) in Experiment 2 for the analysis.”

(iv) You are writing that your participants were students. However, you did mention that one of your inclusion criteria was age till 60 yrs. Is it not confusing? Please carefully address these issues in your revision.

Response: We are sorry if this point is confusing. One of the inclusion criteria for our behavioral experiments is an age between 18 and 60 years. As the participants were recruited through printed advertisements on notice boards at various sites in the EPFL campus, they were actually all student at the EPFL, but it wasn’t formally a inclusion criteria. The mean age was 27.19 years (with a 34 yo oldest subject) in Experiment 1 and 24.04 years (with a 29 yo oldest subject) in Experiment 2. 

We’ve now removed the sentence stating that they were students.

---

## [Editor Report · Decision Letter 2]

6 Dec 2019

PONE-D-19-17841R2

Somatosensory-visual effects in visual biological motion perception

PLOS ONE

Dear Dr Progin:

Thank you for submitting your manuscript to PLOS ONE. After careful consideration, we feel that it has merit but does not fully meet PLOS ONE’s publication criteria as it currently stands. Therefore, we invite you to submit a revised version of the manuscript that addresses the points raised during the review process.

I now had a possibility to look at your reply. Thank you for your efforts. I must admit that there is a problem, which requires youir attention: you wrote that whereas there were no gender differences in Experiment 1 (please indicate whether you did check the data sets for normality of distribution, if the data is not normally distributed, yoiu are unable to use parametric statistics such as t-Student), in Experiment 2 the gender differences are significant (p < 0.03).[Please indicate whether one-tailed or two-tailed statistics is used]. In this latter case, you are unable to consider the whole group consisting of femakes and makles as homogenious. Instead, you have to proceed with the data of females and males separately. I am very sorry, but you really have to address these important issues.  The other issue is exclusion of subjects: you have to explain why so many participants made false alarms. Were they anxious? Was the task extremely demanding?

We would appreciate receiving your revised manuscript by Jan 20 2020 11:59PM. To enhance the reproducibility of your results, we recommend that if applicable you deposit your laboratory protocols in protocols.io, where a protocol can be assigned its own identifier (DOI) such that it can be cited independently in the future. For instructions see: http://journals.plos.org/plosone/s/submission-guidelines#loc-laboratory-protocols

We look forward to receiving your revised manuscript.

Kind regards,

Marina A. Pavlova, PhD

Academic Editor

PLOS ONE

---

## [Author Response · Author response to Decision Letter 2]

9 Jan 2020

RESPONSE TO EDITOR

Dear Dr. Pavlova,

We would like to thank you for further assessing our manuscript, and apologize for the delayed response due to the holiday break.

Here is a response to your comments and concerns. 

Comments to the Authors: I now had a possibility to look at your reply. Thank you for your efforts. I must admit that there is a problem, which requires youir attention: you wrote that whereas there were no gender differences in Experiment 1 (please indicate whether you did check the data sets for normality of distribution, if the data is not normally distributed, yoiu are unable to use parametric statistics such as t-Student), in Experiment 2 the gender differences are significant (p < 0.03).[Please indicate whether one-tailed or two-tailed statistics is used]. In this latter case, you are unable to consider the whole group consisting of femakes and makles as homogenious. Instead, you have to proceed with the data of females and males separately. I am very sorry, but you really have to address these important issues. The other issue is exclusion of subjects: you have to explain why so many participants made false alarms. Were they anxious? Was the task extremely demanding?

Response: We indeed had verified that our data was normally distributed with a Shapiro-Wilk normality test. This was the case both in experiment 1 (W = 0.96, p = 0.67) and in experiment 2 (W = 0.96, p = 0.56). Following standard guidelines, and because we had no a priori hypothesis concerning the influence of gender in our study, we used a two-tailed test to compare a posteriori performance between males and females. 

Instead of proceeding with the data of females and males separately in experiment 2, we added this as a covariate in our analysis. We consider this to be a better approach, as it avoids losing statistical power as would be the case with data splitting. Doing so, the mixed-effects logistic regression yielded essentially similar results as the ones we reported originally. Most importantly, the interaction between stimulus intensity and synchrony remained significant (X² = 11.05, p < 0.001). We added this important point to the revised manuscript. 

Finally, regarding subjects exclusion, we now report in the revised manuscript that some participants had to be excluded because of attention deficit during the task, likely due to the length of the experiment (180 repetitions of PLWs pairs in experiment 1 and 240 repetitions of PLWs pairs in experiment 2). To our knowledge, the task did not trigger anxiety, but was indeed quite demanding. 

We thank you once again for your thorough editorial work, and hope that our study is now suitable for publication in PLOS ONE. 

Best regards, 

Pierre Progin, on behalf of all coauthors.

---

## [Decision Letter · Decision Letter 3]

13 Mar 2020

PONE-D-19-17841R3

Somatosensory-visual effects in visual biological motion perception

PLOS ONE

Dear Dr. Progin:

Thank you for submitting your manuscript to PLOS ONE. After careful consideration, we feel that it has merit but does not fully meet PLOS ONE’s publication criteria as it currently stands. Therefore, we invite you to submit a revised version of the manuscript that addresses the points raised during the review process.

We received feedback from Reviewer 3 (you can find her/his comments below). I believe that two issues are of importance: 1. Stricktly speaking your groups are imbalannced in regard to gender. Experiment 1 contained 8 female and 22 male subjects, Experiment 2 contains 21 male and 7 femnale participants. Binomial test is of no help here. If you have such an imbalanced design, your statisticakl outcome can be affected heavily.  2. The difference between female and male participanrts WAS significant. Significant difference can't be negligible. Please address these issues in your work.

We would appreciate receiving your revised manuscript by Apr 27 2020 11:59PM. To enhance the reproducibility of your results, we recommend that if applicable you deposit your laboratory protocols in protocols.io, where a protocol can be assigned its own identifier (DOI) such that it can be cited independently in the future. For instructions see: http://journals.plos.org/plosone/s/submission-guidelines#loc-laboratory-protocols

We look forward to receiving your revised manuscript.

Kind regards,

Marina A. Pavlova, PhD

Academic Editor

PLOS ONE

Reviewers' comments:

Reviewer's Responses to Questions

**Comments to the Author**

1. If the authors have adequately addressed your comments raised in a previous round of review and you feel that this manuscript is now acceptable for publication, you may indicate that here to bypass the “Comments to the Author” section, enter your conflict of interest statement in the “Confidential to Editor” section, and submit your "Accept" recommendation.

Reviewer #3: All comments have been addressed

2. Is the manuscript technically sound, and do the data support the conclusions?

Reviewer #3: Yes

3. Has the statistical analysis been performed appropriately and rigorously? 

Reviewer #3: Yes

4. Have the authors made all data underlying the findings in their manuscript fully available?

Reviewer #3: Yes

5. Is the manuscript presented in an intelligible fashion and written in standard English?

Reviewer #3: Yes

6. Review Comments to the Author

Reviewer #3: Your paper is clean and easy to read but I think it needs another revision step.

I have a few concerns that follow. To sum them up [1] we need to pay attention to variables operationalization and correspondent stimuli effectiveness for cross modality stimulation (mirror neuron like integration vs higher order cognitive-attentional convergent information about a variable); [2] it is not clear the gender effect; we should [3] pay attention to minor language problems.

I won't use locutions as "in my opinion" etc. because tautological. I apologize for any unpoliteness that should be perceived.

[1] The effectiveness of the asynchronous condition depends on the following fact.

The stimulus in its simplest representation can be described in terms of frequency and phase. The asynchronous condition has to be both different in "frequency" and "phase", otherwise we cannot discard the simplest interpretation that subjects use a low level property of perception (frequency) to give their responses.

I try to be clearer.

The stimulus PLW can be described in terms of frequency of an event ("the [perceived] foot touches the ground"): the higher the frequency, the higher the psychophysical variable we want to measure (the [perceived] velocity of the [perceived] walker). In the synchronous condition there is another stimulus (processed in another perceptual modality) which matches with plw both in phase and in frequency (or at least - please clarify this point - with a frequency which is an integer multiple or divisor of the frequency of the plw event). For example:

SAME PHASE - SAME FREQUENCY

time plw foot stimulus

1 1 1

2 0 0

3 0 0

4 1 1

5 0 0

6 0 0

7 1 1

8 0 0

9 0 0

...

DIFFERENT PHASE - SAME FREQUENCY

time plw foot stimulus

1 1 0

2 0 1

3 0 0

4 0 0

5 1 0

6 0 1

7 0 0

8 0 0

9 1 0

10 0 1

11 0 0

12 0 0

...

SAME PHASE - DIFFERENT FREQUENCY (but integer multiple: frequency foot touch = 2 * frequency plw)

time plw foot stimulus

1 1 1

2 0 0

3 0 1

4 0 0

5 1 1

6 0 0

7 0 1

8 0 0

9 1 1

10 0 0

11 0 1

12 0 0

...

ecc.

The point is: the asynchronous condition should be different both in phase and in frequency with no frequency at all preferably (this can be set giving a random order of the "1's" in the event representation, but maintaining that the total number of "ones" is the same for both stimuli. In analytic terms the two [continuous] event (y) - time (x) functions should have the same integral but one should be a sine(x+shift) and the other some random thing with the same area.

Please add some further consideration about the effectiveness of your "somatosensory drum effect" (some tips about implicit questions you could ask again yourself):

- how can I distinguish the pure "rythm" attentive effect from the "cross modality - mirror neuron like one"? Is my Asynchronous condition enough as it stands? etc.

- How long does each PLW stimulus last?

- is it necessary to refer to the [complex and not universally accepted] theory of mirror neurons or the results can be [easily] explained by other (simpler) mechanisms? In other words if we cannot exclude a simpler process of converging information (different stimuli of which frequencies correlates and help the performance which has a "computation time" compatible with higher order processes), the discussion, say, of rows 461 - 499 has to be "mitigated". Could some further considerations reagarding the comparison of results for forearm vs foot stimulation help? Should this considerations refer also to possible different sensitiveness of those areas (example rows 311-312)?

[2] balanced design are always preferable; your design is not regarding gender and it's unbalanced from the very beginning (it is not only a matter of subjects exclusion for data or inclusion problem). I would not mention at all gender variable effect. If you are somehow obliged to, I'd rather be more explicit about the reasons that make compatible the following statements:

rows 415-417: gender variable has a main effect in experiment 2 and does not have it in experiment 1.

rows: 420-422: the effect of intensity x synchrony remain significant on the dependent variable (the fuzzy-step|sygmoid|error function like decision function) if in the model you add gender variable explicitly.

For what I can understand: on one hand, on its own, if you use t-test, gender has no effect in experiment 1 but has an effect in experiment 2; on the other hand, it does not "significantly modulate" the combined effect of stimulus (plw) intensity and synchrony (with somatosensory stimulation). Those results are compatible without the need of giving any further information on experiment 1; on experiment 2 we are lead to think that (a) the "effect size" of intensity x synchrony is high enough to "cover" the one of gender on its own or [inclusive] (b) that the interaction gender x intensity x synchrony is "small" if significant. If this representation of what you mean in rows 414-424 is correct, please give some statistical effect size measure and be more analytic in its explanation, above all whether if the above a, b or both are true.

In layman terms you should represent you results in order to explain both (1) why gender has effect in experiment 2 and not in experiment 1 and (2) why gender does not affect the intensity x synchrony effect.

Maybe it could be helpful to "let us see" the model. I guess it is something like this:

to each discrete independent variable V_k with number of levels N_k we associate N_k variables X_h with values 0 or 1.

to each numerical/continuous independent variable W_i we associate a numerical variable X_i with the same domain.

So from

[subject, intensity, synchrony, gender]

we obtain

x_1, ..., x_n, with n = number_of_subjects

x_n+1 =intensity

x_n+2 = asynchronous_stimulation

x_n+3 = synchronous_stimulation

x_n+4 = female

x_n+5 = male

and the model WITHOUT interaction is

y = sum(a_j * x_j) + error

For the one with interaction you add more variable such as

x_n+6 = (x_n+1)*(x_n+2 == 1) that is intensity for asynchronous stimulation and so on.

If my rough resume is somehow correct, it should be the case that

y = ... + a*(intensity_x_synchrony) + b*(intensity_x_synchrony_x_gender) + c*(gender) + ...

gives

a significant and strong

b significant and small (or not significant)

c significant (for experiment 2) and small (comparing to a)

am I correct? Nevertheless I suggest you try to find a way to make it clearer.

[3] Miscellanea and minor concerns

Row 244: "...was randomized between each body part ...": check for "between" as it's used for a set of two elements, so it should be "among", because I guess that the body parts are more than 2. Moreover I think you may want something like "... randomized across ... " or "... randomly sampled among all body parts ...". However, if by "body parts" you do not mean those related to moving dots but those of the experimental subject, please be more clear about why you should randomize "left and right" stimulation (I guess because you have only one tool). Nevertheless please check this part with a native speaker.

Row 414: "Although ... NOT ..., the influence is worth considering". I cannot understand this sentence. "Although B, A" means that "A" usually implies "not-B"; in other words "A although B" should be something like "A even if not-B".

A="the influence of gender on BM perception is worth considering"

B="our samples were not significantly unbalanced for gender"="our samples were balanced enough for gender"

so to maintain an intuitive semantics it should be "A because B". Please clarify this point.

Row 450: "STS, with its central [...] actions (see [...]) has been implicated [...]" should be

"STS, with its central [...] actions (see [...]), has been implicated [...]" (the comma is missing). Please double check the paper because there are other punctuation errors.

7. PLOS authors have the option to publish the peer review history of their article (what does this mean?). If published, this will include your full peer review and any attached files.

Reviewer #3: No

---

## [Author Response · Author response to Decision Letter 3]

13 May 2020

Pierre Progin Mai 12th, 2020

Laboratory of Cognitive Neuroscience

Swiss Federal Institute of Technology 

Campus Biotech H4

Chemin des Mines 9

CH-1202 Genève

Switzerland 

RESPONSE TO REVIEWERS

Dear Dr. Pavlova,

Thank you for providing us your valuable feedback on our manuscript titled “Somatosensory-visual effects in visual biological motion perception”. We appreciate the time and effort that you and the reviewer have dedicated to our manuscript and are grateful to the reviewer for her/his insightful comments on our paper. We have been able to incorporate changes to reflect most of the suggestions provided by the reviewers. We have highlighted the changes within the manuscript.

Here is a point-by-point response to the reviewer’s comments and concerns. Please, find below the reviewer’s comments and our responses inserted after each comment.

We thank you once again for your thorough editorial work, and hope that our study is now suitable for publication in PLOS ONE. 

Best regards, 

Pierre Progin, on behalf of all coauthors.

EDITOR’S COMMENTS TO AUTHOR:

Comments to the Author: We received feedback from Reviewer 3 (you can find her/his comments below). I believe that two issues are of importance: 

1. Stricktly speaking your groups are imbalannced in regard to gender. Experiment 1 contained 8 female and 22 male subjects, Experiment 2 contains 21 male and 7 femnale participants. Binomial test is of no help here. If you have such an imbalanced design, your statisticakl outcome can be affected heavily. 

2. The difference between female and male participanrts WAS significant. Significant difference can't be negligible. Please address these issues in your work.

Response: We now make the following statement as an important limitation in the discussion: 

“[...] our sample of participants was not balanced for gender. Only 8 women of 22 subjects participated in Experiment 1 and 7 women of 21 subjects in Experiment 2. Previous studies have shown differences in BM perception depending on gender, showing for instance that female and male observers were found to judge differently whether a frontal PLW is facing toward or backward them (39), as well as emotional expressions (40). Of note, post-hoc analysis revealed better performance in PLW discrimination in male observers in Experiment 2 (77.1% vs. 74.4% t(10.05) = 2.48, p = 0.03), but not in Experiment 1 (t(12.19) = 1.41, p = 0.18). A supplementary binomial mixed-effects regression including gender as a covariate in Experiment 2 revealed that the interaction between intensity and synchrony remained significant (X² = 11.41, p < 0.001), and was not modulated by gender (X² = 0.66, p = 0.41). Future studies will be required to assess whether this effect is anecdotal or whether cross-modal effects on motion processing depends on gender”.

REVIEWER’S COMMENTS TO AUTHOR:

Comments from Reviewer #3: I have a few concerns that follow. To sum them up [1] we need to pay attention to variables operationalization and correspondent stimuli effectiveness for cross modality stimulation (mirror neuron like integration vs higher order cognitive-attentional convergent information about a variable); [2] it is not clear the gender effect; we should [3] pay attention to minor language problems. I won't use locutions as "in my opinion" etc. because tautological. I apologize for any unpoliteness that should be perceived.

[1] The effectiveness of the asynchronous condition depends on the following fact. The stimulus in its simplest representation can be described in terms of frequency and phase. The asynchronous condition has to be both different in "frequency" and "phase", otherwise we cannot discard the simplest interpretation that subjects use a low level property of perception (frequency) to give their responses. I try to be clearer.

The stimulus PLW can be described in terms of frequency of an event ("the [perceived] foot touches the ground"): the higher the frequency, the higher the psychophysical variable we want to measure (the [perceived] velocity of the [perceived] walker). In the synchronous condition there is another stimulus (processed in another perceptual modality) which matches with plw both in phase and in frequency (or at least - please clarify this point - with a frequency which is an integer multiple or divisor of the frequency of the plw event). 

For example:

SAME PHASE - SAME FREQUENCY

time plw foot stimulus

1 1 1

2 0 0

3 0 0

4 1 1

5 0 0

6 0 0

7 1 1

8 0 0

9 0 0

...

DIFFERENT PHASE - SAME FREQUENCY

time plw foot stimulus

1 1 0

2 0 1

3 0 0

4 0 0

5 1 0

6 0 1

7 0 0

8 0 0

9 1 0

10 0 1

11 0 0

12 0 0

...

SAME PHASE - DIFFERENT FREQUENCY (but integer multiple: frequency foot touch = 2 * frequency plw)

time plw foot stimulus

1 1 1

2 0 0

3 0 1

4 0 0

5 1 1

6 0 0

7 0 1

8 0 0

9 1 1

10 0 0

11 0 1

12 0 0

...

ecc.

The point is: the asynchronous condition should be different both in phase and in frequency with no frequency at all preferably (this can be set giving a random order of the "1's" in the event representation, but maintaining that the total number of "ones" is the same for both stimuli. In analytic terms the two [continuous] event (y) - time (x) functions should have the same integral but one should be a sine(x+shift) and the other some random thing with the same area

Response: We thank the reviewer for noting this important aspect and we apologize if our description of the tactile stimuli were not clear enough in the original manuscript. 

Actually, in the synchronous stimulation, the visual stimuli and the tactile stimuli had the same frequency and phase. The visual stimuli consisted of two stimuli, PLW 1 and PLW 2, separated by 1500 ms intervals. The gait frequency was 0.77 cycles per second for PLW 1 and (0.77 + �f cycle/sec) for PLW 2 according to the participants performance (staircase adaptative procedure). PLW 1 or PLW 2 could be presented with equal probability as the first visual stimulus for each trial. The duration of the tactile stimuli was adapted depending on the duration of the visual stimulus (i. e. if the PLW walked faster, the tactile stimulation was shorter) and corresponded to 20% of the duration of the total PLW gait. Indeed, for a full phase of a walking cycle, Foot 1 was stimulated between 10-30% of the visual stimuli cycle and Foot 2 between 60-80% of this cycle. If the cycle was longer/shorter, these stimulation windows were adjusted accordingly. 

To complete your example, we can schematize this as (in row “PLW” 1 indicated that the feet of the PLW touch the ground, in rows “Foot” 1 indicated that tactile stimulations are activated):

SAME PHASE - SAME FREQUENCY

Time PLW1 Foot1 Foot2

1 0 0 0

2 1 1 0

3 1 1 0

4 0 0 0

5 0 0 0

6 0 0 0

7 1 0 1

8 1 0 1

9 0 0 0

10 0 0 0

Interval 1500 ms

Time PLW2 Foot1 Foot2

11 0 0 0

12 1 1 0

13 1 1 0

14 0 0 0

15 0 0 0

16 0 0 0

17 1 0 1

18 1 0 1

19 0 0 0

20 0 0 0

In the asynchronous condition, both frequency and phases were different for the tactile and visual stimuli. The duration of the tactile stimulation was the same as in the synchronous condition (corresponding of 20% of the total PLW gait cycle), but for each foot the phase was randomly selected. For example, this could result in Foot 1 being stimulated from 0-20% of the cycle and Foot 2 from 15-35%. The phase was randomized for each foot separately, i. e. it could also be that Foot 1 and Foot 2 were stimulated at the same time.

For example, it could be like this:

DIFFERENT PHASE – DIFFERENT FREQUENCY

Time PLW1 Foot1 Foot2

1 0 1 0

2 1 1 1

3 1 0 1

4 0 0 0

5 0 0 0

6 0 0 0

7 1 0 0

8 1 0 0

9 0 0 0

10 0 0 0

Interval 1500 ms

Time PLW2 Foot1 Foot2

11 0 0 0

12 1 0 0

13 1 0 0

14 0 1 0

15 0 1 0

16 0 0 0

17 1 0 0

18 1 0 1

19 0 0 1

20 0 0 0

We added these important methodological details in the Method section:

“The duration of the tactile stimuli was adapted depending on the duration of the visual stimulus (i.e. when PLWs walked faster, tactile stimulations were shorter) and corresponded to 20% of the duration of the total PLW gait. 

In the ‘synchronous’ conditions of Experiment 1 and 2, stimulation onset of the laterally matched body part was timed to coincide with the point at which the PLW ‘ankle’ dot reached its lowest value on the y-axis. Indeed, for a full phase of PLW cycle, Foot 1 was stimulated between 10-30% of the visual stimuli cycle and Foot 2 between 60-80% of this cycle.”

“In the ‘asynchronous’ condition of Experiment 2, tactile stimulation onset was randomized during the visual stimulus. Both frequency and phase were different for the visual stimuli and the tactile stimuli. The duration of the tactile stimulation was the same as in the synchronous condition (corresponding of 20% of the total PLW gait cycle), but for each foot the phase was randomly selected. For example, this could result in Foot 1 being stimulated from 0-20% of the cycle and Foot 2 from 15-35%. The phase was randomized for each foot separately, i. e. it could also be that Foot 1 and Foot 2 were stimulated at the same time.

Please add some further consideration about the effectiveness of your "somatosensory drum effect" (some tips about implicit questions you could ask again yourself):

- how can I distinguish the pure "rythm" attentive effect from the "cross modality - mirror neuron like one"? Is my Asynchronous condition enough as it stands? etc.

Response: For Experiment 1, we consider unlikely that the effect comes from a “drum” or “rhythm” effect, as it was not present when the same tactile stimulation was delivered on the forearms. The experimental tactile condition (ET) and control tactile condition (CT) contained indeed the same visual and tactile information about the speed of the PLW. 

We now add this to the Discussion:

“As the frequency and the phase of both visual and tactile stimuli were the same in this experiment, the tactile input itself (through its frequency and its duration) could have conveyed additional information about the speed of the PLW. It is then possible that the improvement in BM discriminability was only due to this additional information. But the finding that the same tactile stimuli improved BM discrimination only when they were delivered under the feet makes this explanation unlikely. There was indeed no difference in BM discrimination when no tactile stimulation occured (V) and when tactile stimulation was applied on the forearms (CT).”

For Experiment 2, as the frequency and the phase are different for the visual and tactile stimuli in the asynchronous condition, it is possible that the better speed discrimination in the synchronous condition comes from such “rhythm” matching between the two sensory inputs. 

We now discuss this possibility in the Discussion:

“However, as the frequency and the phase are different for the visual and tactile stimuli in the asynchronous condition, we cannot exclude, based only on Experiment 2, that the improvement in BM discriminability in the synchronous condition comes from additional information about the speed of the PLW conveyed by the tactile stimulation itself.”

- How long does each PLW stimulus last?

Response: We used the monitor refresh rate as the reference (100Hz) and each frame showed a different dot pattern. The gait frequency of 0.77 cycles per second was used as the base gate frequency, meaning that 129 frames were shown for a full PLW cycle, lasting 1298 ms. An interval of 1500 ms separated the presentation of this reference PLW to a second PLW with a slightly faster gait frequency (0.77 + �f cycle/sec) according to the participants performance (staircase adaptative procedure). For instance, if during the previous trials the discrimination was incorrect with a �f of 0.2 cycle/sec, then the second walker would have 100/(0.77+0.2) = 103 frames for a full cycle, lasting 1030 ms. Then the total duration varied for each trial, with a maximum of 4096 ms (1298 + 1500 + 1298 ms).

- is it necessary to refer to the [complex and not universally accepted] theory of mirror neurons or the results can be [easily] explained by other (simpler) mechanisms? 

In other words if we cannot exclude a simpler process of converging information (different stimuli of which frequencies correlates and help the performance which has a "computation time" compatible with higher order processes), the discussion, say, of rows 461 - 499 has to be "mitigated". Could some further considerations regarding the comparison of results for forearm vs foot stimulation help? 

Response: Following the Reviewer’s suggestion, we have left the reference to mirror neurons, which was not the aim of this study, and now also propose alternative hypotheses regarding the difference of results for forearm and feet stimulation in the Discussion section (i.e., difference in sensitiveness between the two body parts, higher relevance of the feet for an embodied representation of walking actions).

Should this considerations refer also to possible different sensitiveness of those areas (example rows 311-312)?

Response: The sensitivity of the tactile simulation on the arm was tested in a pilot study. Moreover, participants reported strong tactile sensations during the task. Accordingly, we argue that it is unlikely that performance difference in Experiment 1 are due to a difference in sensitiveness between feet and forearms, although we cannot exclude this. We now have mentioned this limitation in the revised Discussion:

“Secondly, we cannot exclude the possibility that differences in tactile perception between the forearms and the feet may have resulted from difference in sensitiveness between these two body parts. We hypothesized during the study conception that the feet would be more relevant than forearms regarding walking actions, and therefore predicted that performance would be better with tactile stimulation on this site than elsewhere on the body.”

[2] balanced design are always preferable; your design is not regarding gender and it's unbalanced from the very beginning (it is not only a matter of subjects exclusion for data or inclusion problem). I would not mention at all gender variable effect. 

Response: We share the reviewer’s opinion, but decided to follow the editor’s suggestion and report post-hoc effects on gender (see above). 

If you are somehow obliged to, I'd rather be more explicit about the reasons that make compatible the following statements:

rows 415-417: gender variable has a main effect in experiment 2 and does not have it in experiment 1.

rows: 420-422: the effect of intensity x synchrony remain significant on the dependent variable (the fuzzy-step|sygmoid|error function like decision function) if in the model you add gender variable explicitly.

For what I can understand: on one hand, on its own, if you use t-test, gender has no effect in experiment 1 but has an effect in experiment 2; on the other hand, it does not "significantly modulate" the combined effect of stimulus (plw) intensity and synchrony (with somatosensory stimulation). Those results are compatible without the need of giving any further information on experiment 1; on experiment 2 we are lead to think that (a) the "effect size" of intensity x synchrony is high enough to "cover" the one of gender on its own or [inclusive] (b) that the interaction gender x intensity x synchrony is "small" if significant. If this representation of what you mean in rows 414-424 is correct, please give some statistical effect size measure and be more analytic in its explanation, above all whether if the above a, b or both are true.

Response: Following the reviewer’s suggestion, we conducted a supplementary binomial mixed-effects regression including gender as a covariate of (no) interest in Experiment 2. We are pleased to report that the interaction between intensity and synchrony remained significant (X² = 11.41, p < 0.001), and was not modulated by gender (X² = 0.66, p = 0.41). 

In layman terms you should represent you results in order to explain both (1) why gender has effect in experiment 2 and not in experiment 1 and (2) why gender does not affect the intensity x synchrony effect.

Maybe it could be helpful to "let us see" the model. I guess it is something like this:

to each discrete independent variable V_k with number of levels N_k we associate N_k variables X_h with values 0 or 1.

to each numerical/continuous independent variable W_i we associate a numerical variable X_i with the same domain.

So from

[subject, intensity, synchrony, gender]

we obtain

x_1, ..., x_n, with n = number_of_subjects

x_n+1 =intensity

x_n+2 = asynchronous_stimulation

x_n+3 = synchronous_stimulation

x_n+4 = female

x_n+5 = male

and the model WITHOUT interaction is

y = sum(a_j * x_j) + error

For the one with interaction you add more variable such as

x_n+6 = (x_n+1)*(x_n+2 == 1) that is intensity for asynchronous stimulation and so on.

If my rough resume is somehow correct, it should be the case that

y = ... + a*(intensity_x_synchrony) + b*(intensity_x_synchrony_x_gender) + c*(gender) + ...

gives

a significant and strong

b significant and small (or not significant)

c significant (for experiment 2) and small (comparing to a)

am I correct? Nevertheless I suggest you try to find a way to make it clearer

Response: The reviewer is correct. We provide below the table corresponding to the analysis of Deviance (Type II Wald chi square tests) of the model: resp ~ intensity* synchrony * orientation * sex 

--Please see the "response to reviewers_12.05.20.docx" document to access the table--

[3] Miscellanea and minor concerns

Row 244: "...was randomized between each body part ...": check for "between" as it's used for a set of two elements, so it should be "among", because I guess that the body parts are more than 2. Moreover I think you may want something like "... randomized across ... " or "... randomly sampled among all body parts ...". However, if by "body parts" you do not mean those related to moving dots but those of the experimental subject, please be more clear about why you should randomize "left and right" stimulation (I guess because you have only one tool). Nevertheless please check this part with a native speaker.

Response: We would like to thank the reviewer for this important comment. We meant among each body parts. We now give more details about the tactile stimuli in the Apparatus and Stimuli section:

“In the ‘asynchronous’ condition of Experiment 2, tactile stimulation onset was randomized during the visual stimulus. Both frequency and phase were different for the visual stimuli and the tactile stimuli. The duration of the tactile stimulation was the same as in the synchronous condition (corresponding of 20% of the total PLW gait cycle), but for each foot the phase was randomly selected. For example, this could result in Foot 1 being stimulated from 0-20% of the cycle and Foot 2 from 15-35%. The phase was randomized for each foot separately, i. e. it could also be that Foot 1 and Foot 2 were stimulated at the same time.” 

Row 414: "Although ... NOT ..., the influence is worth considering". I cannot understand this sentence. "Although B, A" means that "A" usually implies "not-B"; in other words "A although B" should be something like "A even if not-B".

A="the influence of gender on BM perception is worth considering"

B="our samples were not significantly unbalanced for gender"="our samples were balanced enough for gender"

so to maintain an intuitive semantics it should be "A because B". Please clarify this point.

Response: The reviewer is right, this sentence was incoherent. Actually there was a mistake in the first part of the sentence, it should have been: “although our samples were not significantly balanced for gender �…�” and not “�…� unbalanced �…�”. We don’t use this sentence anymore in the manuscript.

Row 450: "STS, with its central [...] actions (see [...]) has been implicated [...]" should be

"STS, with its central [...] actions (see [...]), has been implicated [...]" (the comma is missing). Please double check the paper because there are other punctuation errors.

Response: We thank the reviewer for these observations. We have double checked the revised manuscript for punctuation errors.

---

## [Editor Report · Decision Letter 4]

19 May 2020

Somatosensory-visual effects in visual biological motion perception

PONE-D-19-17841R4

Dear Dr. Progin,

We are pleased to inform you that your manuscript has been judged scientifically suitable for publication and will be formally accepted for publication once it complies with all outstanding technical requirements.

With kind regards,

Marina A. Pavlova, PhD

Academic Editor

PLOS ONE
---

## [Editor Report · Acceptance letter]

29 May 2020

PONE-D-19-17841R4 

Somatosensory-visual effects in visual biological motion perception 

Dear Dr. Progin:

I am pleased to inform you that your manuscript has been deemed suitable for publication in PLOS ONE. Congratulations! Your manuscript is now with our production department. 

With kind regards,

on behalf of

Prof. Marina A. Pavlova 

Academic Editor

PLOS ONE